# Sustainable Self-evolution Adversarial Training

## ABSTRACT

With the wide application of deep neural network models in various computer vision tasks, there has been a proliferation of adversarial example generation strategies aimed at exploring model security deeply. However, existing adversarial training defense models, which rely on single or limited types of attacks under a one-time learning process, struggle to adapt to the dynamic and evolving nature of attack methods. Therefore, to achieve defense performance improvements for models in long-term applications, we propose a novel Sustainable Self-evolution Adversarial Training (SSEAT) framework. Specifically, we introduce a continual adversarial defense pipeline to realize learning from various kinds of adversarial examples across multiple stages. Additionally, to address the issue of model catastrophic forgetting caused by continual learning from ongoing novel attacks, we propose an adversarial data replay module to better select more diverse and key relearning data. Furthermore, we design a consistency regularization strategy to encourage current defense models to learn more from previously trained ones, guiding them to retain more past knowledge and maintain accuracy on clean samples. Extensive experiments have been conducted to verify the efficacy of the proposed SSEAT defense method, which demonstrates superior defense performance and classification accuracy compared to competitors.

## CCS CONCEPTS

• **Do Not Use This Code** → **Generate the Correct Terms for Your Paper**; *Generate the Correct Terms for Your Paper*; Generate the Correct Terms for Your Paper; Generate the Correct Terms for Your Paper.

## KEYWORDS

Adversarial Training, Model Defense, Adversarial Examples, Continue Learning

## 1 INTRODUCTION

Deep learning has been widely applied in computer vision tasks such as image classification [8, 65] and object detection [6, 37], resulting in significant advancements. However, the vulnerability of deep learning models to adversarial attacks [16, 26, 49] has become a critical concern. Adversarial examples involve intentionally crafted small perturbations that deceive deep learning models, leading them to produce incorrect outputs. This poses a serious threat to the reliability and security of these models in real-world applications.

*ACM MM, 2024, Melbourne, Australia*
© 2024 Copyright held by the owner/author(s). Publication rights licensed to ACM.
ACM ISBN 978-x-xxxx-xxxx-x/YY/MM
https://doi.org/10.1145/nnnnnnn.nnnnnnn

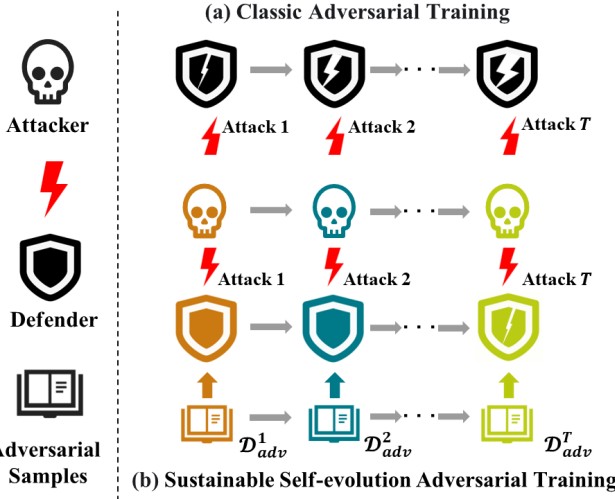

Figure 1: A conceptual overview of our Sustainable Self-evolution Adversarial Training (SSEAT) method. When confronted with the challenge of ongoing generated new adversarial examples in complex and long-term multimedia applications, existing adversarial training methods struggle to adapt to iteratively updated attack methods. In contrast, our SSEAT model achieves sustainable defense performance improvements by continuously absorbing new adversarial knowledge.

Consequently, research on defense mechanisms [17, 22, 34] has become increasingly essential and urgent.

Nowadays, researchers study various model defense methods to address highly destructive adversarial samples, including input sample denoising [21, 24] and attack-aware detection [14, 15]. Among these defense methods, adversarial training [12, 67] stands out as one of the most effective defense strategies. Adversarial training is a game-based training approach aimed at maximizing perturbations while minimizing adversarial expected risk. Its core idea is to integrate generated adversarial examples into the training set, enabling the model to learn from these examples during training and enhance its robustness.

Current adversarial training strategies often rely on one or limited types of adversarial examples to achieve robust learning, and focus on improving the defense performance against attacks and the efficiency of the adversarial training process. However, in real-world applications, as researchers delve deeper into model defense and security, various new attack strategies continue to emerge (According to our incomplete statistics, over the past five years, more than 200 papers on adversarial attack algorithms have been published in top journals and conferences every year across various fields including multimedia, artificial intelligence, and computer

vision), existing adversarial training strategies obviously struggle to address such complex scenario.

Therefore, it is essential for deep models to achieve sustainable improvement in defense performance for long-term application scenarios, as shown in Fig. 1. This has brought about the following challenges: (1) How to achieve the sustainability of the adversarial training strategy when new adversarial examples are constantly being born; (2) How to solve the model catastrophic forgetting problem caused by continuous exposure to new adversarial examples; (3) How to balance the model's robustness on adversarial examples and accuracy on clean data.

In this study, to address the above three challenges for the defense model against the ongoing generation of new adversarial examples, we propose a novel and task-driven Sustainable Self-evolution Adversarial Training (SSEAT) framework to ensure the model maintains its accuracy and possesses robust and continuous defense capabilities. Our SSEAT defense method comprises three components: Continual Adversarial Defense (CAD), Adversarial Data Reply (ADR), and Consistency Regularization Strategy (CRS). To achieve sustainability in the adversarial training strategy (Challenge (1)), drawing inspiration from the continue learning paradigm, we propose a CAD pipeline, which learns from one type of adversarial example at each training stage to address the continual generation of new attack algorithms. To address the issue of catastrophic forgetting when continuously learning from various attacks (Challenge (2)), we introduce an ADR module to establish an effective re-learning sample selection scheme, advised by classification uncertainty and data augmentation. Meanwhile, to realize the trade-off between the model's robustness against adversarial examples and accuracy on clean data (Challenge (3)), We designed a CRS module to help the model not overfit to current attacks and prevent the model from losing knowledge on clean samples. Overall, our SSEAT method effectively addresses a range of defense challenges arising from continuously evolving attack strategies, maintaining high classification accuracy on clean samples, and ensuring lifelong defense performance against ongoing new attacks.

We summarize the main contributions of this paper as follows:

- We recognize the challenges of continuous defense setting, where adversarial training models must adapt to ongoing new kinds of attacks. This deep model defense task is of significant practical importance in real-world applications.
- We propose a novel sustainable self-evolution adversarial training algorithm to tackle the problems under continuous defense settings.
- We introduce a continual adversarial defense pipeline to learn from diverse types of adversarial examples across multiple stages, an adversarial data reply module to alleviate the catastrophic forgetting problem when the model continuously learns from new attacks, and a consistency regularization strategy to prevent significant accuracy drop on clean data.
- Our approach has yielded excellent results, demonstrating robustness against adversarial examples while maintaining high

## 2 RELATED WORK

### 2.1 Adversarial Attacks

The impressive success of deep learning models in computer vision tasks [6, 37, 62] has sparked significant research interest in studying their security. Many researchers are now focusing on adversarial example generation [16, 26, 49]. Adversarial examples add subtle perturbations that are imperceptible to the human eye on clean data, causing the model to produce incorrect results. Depending on the access rights to the target model and data, attacks can be divided into black-box attacks [1, 38, 45, 47] and white-box attacks [4, 19, 40, 46]. Most white-box algorithms [19, 31, 40] obtain adversarial examples based on the gradient of the loss function to the inputs by continuously iteratively updating perturbations. In black-box attacks, some methods [52, 68] involve iteratively querying the outputs of the target model to estimate its gradients by training a substitute model, while others [5, 41, 44] concentrate on enhancing the transferability of adversarial examples between different models. Over time, new algorithms for generating attack examples are continually being developed. Therefore, our focus is on addressing the ongoing creation of new attacks while maintaining the model's robustness against them.

### 2.2 Adversarial Training

Adversarial training [19] is a main method to effectively defend against adversarial attacks. This approach involves augmenting the model's training process by incorporating adversarial examples, thus the data distribution learned by the model includes not only clean samples but also adversarial examples. Many adversarial training research mainly focuses on improving training efficiency and model robustness [11, 32, 57, 67]. For example, Zhao[66] uses FGSM instead of PGD during training to reduce training time and enhance efficiency, Dong [12] investigates the correlation between network structure and robustness to develop more robust network modules, Chen [7] uses data enhancement or generative models to alleviate robust overfitting, and Lyu [39] adopts regularization training strategies, such as stopping early to smooth the input loss landscape. Meanwhile, the trade-off between robustness and accuracy has attracted much attention [34, 43, 51]. TRADES [63] utilizes Kullback-Leibler divergence (KL) loss to drive clean and adversarial samples closer in model output, balancing robustness and accuracy. In addition, some studies [48, 64] try to use curriculum learning strategies to improve robustness while reducing the decrease in accuracy on clean samples. Most current adversarial training approaches rely on a single or limited adversarial example generation algorithm to enhance model robustness. However, in real-world scenarios, existing defense methods struggle to address the ongoing emergence of diverse adversarial attacks. Inspired by the continue learning paradigms, we aim to enhance defense capabilities by enabling the adaptive evolution of adversarial training algorithms for long-term application scenarios.

### 2.3 Continue Learning

Continue learning [2, 28, 58] aims at the model being able to continuously learn new data without forgetting past knowledge. Continue learning can be mainly divided into three categories. One is based

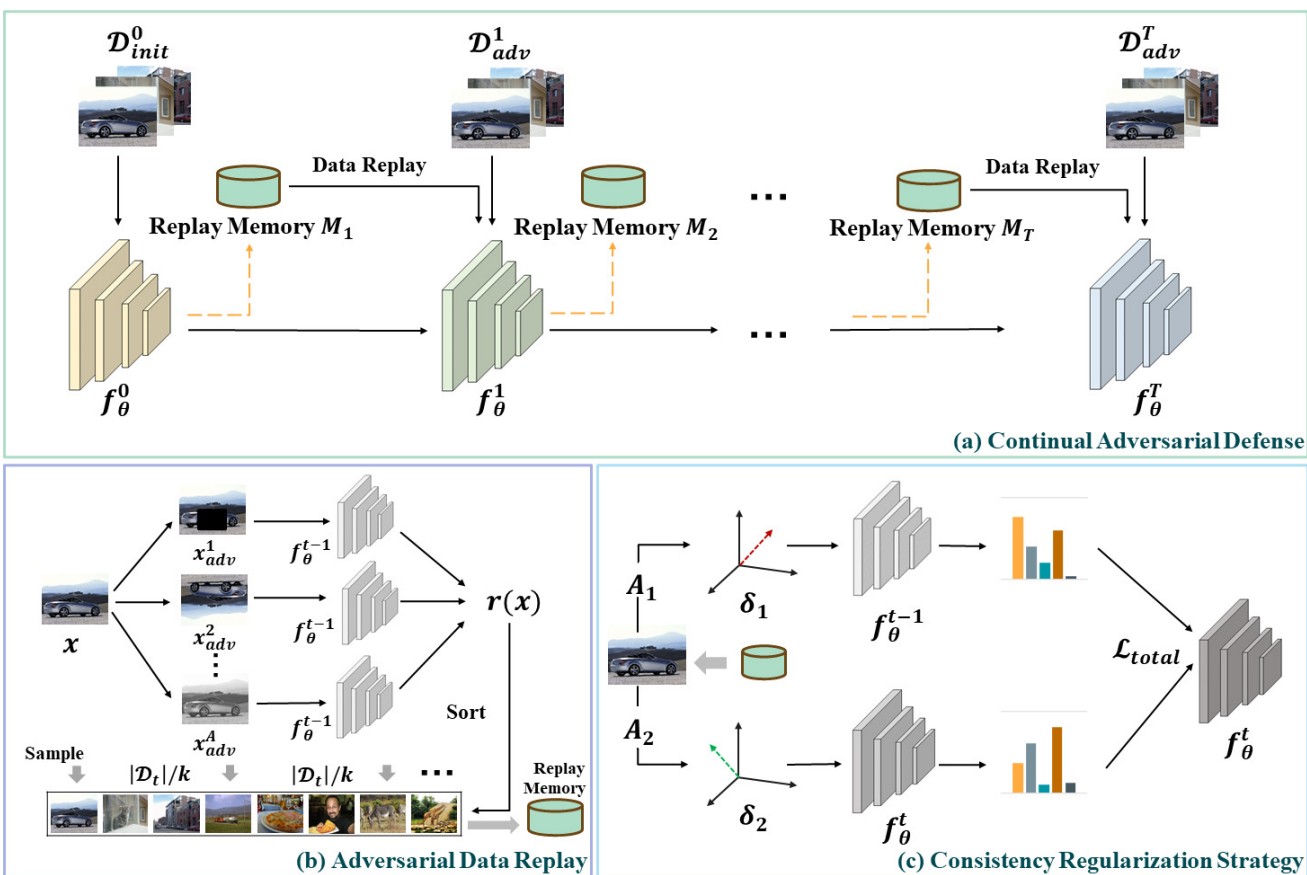

Figure 2: (a) Illustration of our Continual Adversarial Defense (CAD) pipeline. CAD helps the model to learn from new kinds of attacks in multiple stages continuously. (b) Illustration of our Adversarial Data Replay (ADR) module. ADR guides the model to select diverse and representative replay data to alleviate the catastrophic forgetting issue. (c) Illustration of our Consistency Regularization Strategy (CRS) component. CRS encourages the model to learn more from the historically trained models to maintain classification accuracy.

on the regularization of model parameters [27, 61] by preserving important parameters from the past while updating less critical ones. However, this method's performance is not ideal when applied to scenarios involving a large number of tasks. One is based on knowledge playback [3, 9, 35, 42, 50], where important past samples are stored in memory and used for training when encountering new tasks. Another approach involves dynamically expanding model parameters [18, 23, 33] to assign different parameters to different tasks. This method helps alleviate catastrophic forgetting and enhances model performance, but it requires significant memory and computational resources. Nevertheless, continue learning models are also susceptible to adversarial examples. Wang *et. al.* [54] attempt to combine adversarial training with continue learning paradigms, but they do not consider the defensive performance of the current model against old attacks in long-term application scenarios. Differently from previous work, our focus is on ensuring that when the model continues to encounter new adversarial attacks, it can maintain robustness against past adversarial examples and improve resilience against new adversarial attacks.

## 3 METHODOLOGY

### 3.1 Task Defination and Framework Overview

**Continuous Defense Setting.** As deep learning models become widely used in fields like healthcare, manufacturing, and military, ensuring their security has become a primary focus for researchers. Investigating adversarial examples with high transferability provides valuable insights into deep models. Consequently, there has been a constant influx of diverse new attack methods [16, 26, 49] in recent years. To develop defense algorithms suitable for long-term applications, we have designed a new Continuous Defense Setting (CDS). In the CDS, a model trained on clean data needs to continually cope with and learn from newly generated adversarial examples, meanwhile, due to limited storage resources, it is impractical to retain a vast number of learned samples. Therefore, our research focuses on addressing the challenge of catastrophic forgetting, improving the model's defense against various attacks, and maintaining high accuracy on clean samples.

**Framework Overview.** To address the ongoing generation of diverse adversarial samples and tackle model defense challenges in long-term application scenarios, we propose a Sustainable Self-evolution Adversarial Training (SSEAT) algorithm under CDS, containing Continual Adversarial Defense (CAD), Adversarial Data Reply (ADR), and Consistency Regularization Strategy (CRS) three components. As shown in Fig. 2, based on the continue learning paradigm, the CAD employs a min-max adversarial training optimization process to continually learn from new attack samples, as described in Sec. 3.2. To alleviate the catastrophic forgetting issue and boost the model robustness of the diverse attacks, in Sec. 3.3, we introduce a DR module to adaptively select important samples from the previous learning stage for data replay during training. Meanwhile, inspired by the knowledge distillation strategy, in Sec. 3.4, we also design a CRS module to shorten the distance between the current model and the model trained in the previous stage, thereby maintaining the model's recognition performance on clean samples over time.

## 3.2 Continual Adversarial Defense

In a classification task, a dataset $\mathcal{D}$ consists of $n$ pairs $(x_i, y_i)$, where $x_i$ represents input samples and $y_i$ denotes corresponding class labels ranging from integers 1 to $K$. The classification model $f_\theta$ is intended to map the input space $\mathcal{X}$ to the output space $\Delta^{K-1}$, generating probability outputs through a softmax layer. To deal with the boom-growing attack strategies, the concept of adversarial robustness extends beyond evaluating the model's performance solely on $\mathcal{P}$. It involves assessing the model's ability to handle perturbed samples within a certain distance metric range around $\mathcal{P}$. Specifically, our goal is to achieve $l_p - robustness$, where we aim to train a classifier $f_\theta$ to accurately classify samples $(x + \delta, y)$ under any $\delta$ perturbation such that $\|\delta\|_p \leq \epsilon$. Here, $(x, y)$ follows the distribution $\mathcal{P}$, and $p \geq 1$ with a small $\epsilon > 0$.

The core concept of adversarial training is to incorporate generated adversarial examples into the training set, allowing the model to learn from these adversarial examples during training, thereby acquiring more robust features and enhancing the model's defense capability. Adversarial training can be formalized as a min-max optimization problem: the goal is to find model parameters $\theta$ that enable the correct classification of adversarial examples,

$$min_\theta \mathbb{E}_{(x,y) \sim \mathcal{D}} \left[ \max_{\|\delta\|_p \leq \epsilon} \mathcal{L}_{adv}(\theta, x + \delta, y) \right] \quad (1)$$

where $\mathcal{L}_{adv}$ represents the loss function, and we use the standard cross-entropy loss to design the loss function $\mathcal{L}_{adv}$.

In our practical CDS, the adversarial training model will continue to encounter adversarial examples generated in various ways. Thus, we design a novel Continuous Adversarial Defense (CAD) pipeline, at each stage of CAD, the model is exposed to a batch of new attack samples for training to adapt continuously to new environments and data distributions. In the initial stage, the model is trained on original clean data, and the trained model is denoted as $f_{\theta_0}$. Additionally, we generate multiple sets of adversarial examples targeting the same original sample using different attack methods, and each adversarial example set corresponds to a specific attack method. During the CAD, the model trains on adversarial

examples generated by specific attack methods in each learning stage. After each stage, the model is updated and denoted as $f_{\theta_t}$, where $t$ represents the stage number. The training process of the model is described as follows,

**Initial Stage:** The model $f_{\theta_0}$ is trained on the original clean sample set $\mathcal{D}_{init}^0 = \{(x_i, y_i)\}_{i=1}^n$, where $x_i$ is the input sample and $y_i$ is the corresponding label.

**CAD Training Stage:** In the $t_{th}$ stage, the model $f_{\theta_{t-1}}$ receives a set of adversarial samples $\mathcal{D}_{adv}^t = \left\{ \left( x_i^t, y_i^t \right) \right\}_{i=1}^n$ generated by the $t_{th}$ attack method for training, resulting in the updated model $f_{\theta_t}$.

## 3.3 Adversarial Data Reply

During the CAD, as more and more attack examples are incorporated into training, the model increasingly struggles to avoid catastrophic forgetting, hindering its ability to maintain sustainable defense capabilities in long-term application scenarios. Thus, we introduce a novel Adversarial Data Reply (ADR) strategy to realize an effective rehearsal sample selection scheme, enhance adversarial example diversity, and obtain high-quality replay data. High-quality sample data should accurately reflect their class attributes and demonstrate clear distinctions from other classes in the feature space. We consider samples located at the distribution center to be the most representative, while those at the classification boundary are the most distinctive. Therefore, based on these two characteristics, we select diverse and representative replay data within the feature space.

However, accurately computing the relative position of samples in the feature space requires significant computational resources and time. Therefore, we utilize our classification model to infer the uncertainty of samples, thereby indirectly revealing their relative positions in the feature space. In practical implementation, we perform various data augmentations to obtain augmented samples. Subsequently, we calculate the variance of the model's output results for these samples subjected to different data augmentations to assess their uncertainty. We think that when the model's predictions for a sample are more certain, the sample may be closer to the core of the class distribution; conversely, when uncertainty in predictions increases, the sample may be closer to the class boundary.

First, we define the learning samples for each round as,

$$\mathcal{D}_t = \begin{cases} \mathcal{D}_{init}^0, & t = 0 \\ \mathcal{D}_{adv}^t, & t > 0 \end{cases} \quad (2)$$

where $\mathcal{D}_t$ represents the sample set of the $t_{th}$ learning stage in CAD.

We assume that the prior distribution of samples $p(\tilde{x}|x)$ is a uniform mixture of various data augmentations, where $\tilde{x}$ represents the augmented samples generated via color jitter, shear, or cutout. We utilize the Monte Carlo method to approximate the uncertainty of the sample distribution $p(y = c|x)$. Then, we measure the relative distribution of samples based on the uncertainty of model outputs.

The derivation process is as follows,

$$p\left(y = c|x\right) = \int_{\tilde{\mathcal{D}}} p\left(y = c|\tilde{x}_t\right) p\left(\tilde{x}_t|x\right) d\tilde{x}_t$$

$$\approx \frac{1}{Z} \sum_{t=1}^{Z} p\left(y = c|\tilde{x}_t\right) \tag{3}$$

where $Z$ signifies the number of augmentation methods utilized. The distribution $\tilde{D}$ represents the data distribution defined by $\tilde{x}$. $p\left(y = c|\tilde{x}_t\right)$ denotes the probability of the augmented sample $\tilde{x}_t$ having the label $c$.

Specifically, the augmented sample $\tilde{x}$ is generated by a random function $g_k(\cdot)$,

$$\tilde{x} = g_k\left(x, \alpha_k\right), k = 1, ...K \tag{4}$$

where $\alpha_k$ represents a hyper-parameter signifying the stochastic component of the $k_{th}$ perturbation.

The prior distribution $p\left(y = \tilde{x}|x\right)$ is formulated as,

$$\tilde{x} \sim \sum_{k=1}^{K} \omega_k * g_k\left(x, \alpha_k\right) \tag{5}$$

where the random variable $\omega_k$ is selected from a categorical binary distribution. We assess the sample's uncertainty in relation to the perturbation by,

$$Q_c = \sum_{t=1}^{T} \mathcal{W} \underset{c'}{\text{argmax}} \, p\left(y = c'|\tilde{x}_t\right) \tag{6}$$

$$r\left(x\right) = 1 - \frac{1}{A} \underset{c}{max} Q_c \tag{7}$$

where $r\left(x\right)$ represents the uncertainty of sample $x$, $Q_c$ indicates the number of times augmented samples are predicted as the true class, $\mathcal{W}$ represents the one-hot encoded class vector, where only the element corresponding to the true class is true. Lower values of $r\left(x\right)$ indicate that the sample resides closer to the distribution center.

We allocate memory for a replay buffer of size $K$ for each learning iteration. We sort all samples $\mathcal{D}_t$ based on the computed uncertainty $r\left(x\right)$, and sample the examples with an interval of $|\mathcal{D}_t|/K$. We have diversified the replay samples by sampling perturbed samples of varying intensities, ranging from robust to fragile ones, which can broaden the scope of memories, encompassing a wide range of scenarios.

## 3.4 Consistency Regularization Strategy

In practical application models, in addition to achieving sustainable defense against attack examples, it is crucial to maintain high recognition accuracy on original clean samples. To prevent the model's learned data distribution from straying too far from the space of clean sample data, we leverage the knowledge distillation method and propose a novel Consistency Regularization Strategy (CRS), to ensure that the same sample fed into both previous model $f_{\theta_{t-1}}$ and current training model $f_{\theta_t}$, after undergoing independent data augmentations, still yields similar predictions.

For a given training sample $(x, y) \sim \mathcal{D}$ and augmentation $A \sim \mathcal{A}$, the training loss is given by,

$$\underset{\|\delta\|_{\infty} \leq \epsilon}{max} \, \mathcal{L}_{CE}(f_{\theta_t}\left(A\left(x\right) + \delta\right), y) \tag{8}$$

where $\mathcal{A}$ represents the baseline augmentation set. $f_{\theta_t}$ represents the model parameters during the $t_{th}$ rounds of CAD.

Considering data points $((x, y)$ drawn from distribution $\mathcal{D}$, and augmentations $A_1$ and $A_2$ sampled from set $\mathcal{A}$, we denote the adversarial noise of $A_i\left(x\right)$ as $\delta_i$. It is obtained by $\delta_i := argmax_{\|\delta\|_{p \leq \epsilon}} \mathcal{L}\left(A_i\left(x\right), y, \delta; \theta_t\right)$. Our objective is to regularize the temperature-scaled distribution $\hat{f_{\theta_t}}\left(x; \tau\right)$ of adversarial examples across augmentations for consistency. Here, $\tau$ is the temperature hyperparameter. Specifically, we use temperature scaling to adjust the classifier: $\hat{f_{\theta_t}}\left(x; \tau\right) = Softmax\left(\frac{z_{\theta_t}\left(x\right)}{\tau}\right)$, where $z_{\theta_t}\left(x\right)$ is the logit value of $f_{\theta_t}\left(x\right)$ before the softmax operation. Therefore, we obtain the regularization loss as follows:

$$JS\left(\hat{f_{\theta_{t-1}}}\left(A_1\left(x\right) + \delta_1, \tau\right) \, \| \, \hat{f_{\theta_t}}\left(A_2\left(x\right) + \delta_2, \tau\right)\right) \tag{9}$$

where $JS\left(\cdot \| \cdot\right)$ denotes the Jensen-Shannon divergence. Since augmentations are randomly sampled at each training step, minimizing the proposed objective ensures that adversarial examples remain consistently predicted regardless of augmentation selection. Additionally, in adversarial training, due to the relatively low confidence of predictions (i.e., maximum softmax value), using a smaller temperature helps ensure a sharper distribution to address this issue.

By ensuring consistency between the predictions of the previous model and the current training model under different data augmentation schemes, we can ensure that the previous model retains the knowledge learned from the previous training. This CRS approach not only helps improve the model's robustness to adversarial samples but also maintains accuracy on clean samples.

The overarching training objective, denoted as $L_{total}$, integrates adversarial training objectives with consistency regularization losses. Initially, we deliberate on averaging the inner maximization objective $L_{adv}$ across two distinct augmentations, $A_1$ and $A_2$, sampled from the augmentation set $A$. This choice stems from the equivalence of minimizing over the augmentation set $A$ to averaging over $A_1$ and $A_2$.

$$\frac{1}{2}\left(\mathcal{L}_{adv}\left(A_1\left(x\right), y; \theta_{t-1}\right) + \mathcal{L}_{adv}\left(A_2\left(x\right), y; \theta_t\right)\right) \tag{10}$$

Subsequently, we integrate our regularizer into the averaged objective mentioned above, introducing a hyperparameter denoted as $\lambda$. Therefore, the final training objective $L_{total}$ can be expressed as follows:

$$\mathcal{L}_{total} := \frac{1}{2}\left(\mathcal{L}_{adv}\left(A_1\left(x\right), y; \theta_{t-1}\right) + \mathcal{L}_{adv}\left(A_2\left(x\right), y; \theta_t\right)\right)$$
$$+ \lambda \cdot JS\left(\hat{f_{\theta_{t-1}}}\left(A_1\left(x\right) + \delta_1, \tau\right) \, \| \, \hat{f_{\theta_t}}\left(A_2\left(x\right) + \delta_2, \tau\right)\right) \tag{11}$$

Our regularization method stands apart from the selection of the adversarial training objective, rendering it universally applicable to diverse established adversarial training approaches. To illustrate, if we were to adopt the conventional adversarial training loss, the

resulting overall objective would be formulated as:

$$\mathcal{L}_{total} := \frac{1}{2} \left( \max_{\|\delta_1\|_{p \leq \epsilon}} \mathcal{L}_{CE}(f_{\theta_{t-1}}(A_1(x) + \delta_1), y) + \right.$$

$$\left. \max_{\|\delta_2\|_{p \leq \epsilon}} \mathcal{L}_{CE}(f_{\theta_t}(A_2(x) + \delta_2), y) \right)$$

$$+ \lambda \cdot JS \left( \hat{f_{\theta_{t-1}}}(A_1(x) + \delta_1, \tau) \| \hat{f_{\theta_t}}(A_2(x) + \delta_2, \tau) \right) \quad (12)$$

---

**Algorithm 1** Sustainable Self-evolution Adversarial Training.

**Require:**

**Input:** Clean sample set $\mathcal{D}^0_{init}$, Adversarial sample sets $\mathcal{D}^t_{adv}$ for each learning stage $t$, target model $f_\theta$, Augmentation set $A$, Temperature hyperparameter $\tau$, regularization coefficient $\lambda$, training epoch $T$.

**Output:** Defense model $f_{\theta_T}$

**Initialization:** Train the initial model $f_{\theta_0}$ by dataset $\mathcal{D}^0_{init}$, Select replay samples $Dr_1$ according to Eq. (7)

1: **for** $t$ in $1, \cdots, T$ **do**
2:    in $D^t_{adv}$, apply adversarial training, according to Eq. (10), to train the model
3:    in $Dr_t$, apply adversarial training and consistency regularization strategy, according to Eq. (12), to train the model $f_{\theta_t}$
4:    Combine replay samples with adversarial samples of the current stage : $D_t = \left\{ (x, y) \mid, (x, y) \in Dr_t \cup \mathcal{D}^t_{adv} \right\}$, and select replay samples $Dr_{t+1}$ according to Eq. (7)
5: **end for**
6: **return** $f_{\theta_T}$

---

## 4 EXPERIMENTAL

### 4.1 Experimental Settings

**Datasets.** We evaluate our SSEAT model over the CIFAR-10 dataset [29],which is commonly used for adversarial attack and defense research. It contains 50,000 images for training and 10,000 images for testing, covering 10 different categories of objects.Our method only uses 1000 images for training and all 1000 images for testing. In each stage, the training and test data are converted into attack according to the corresponding attack algorithm. The converted training part of the data is used for SSEAT training, and the test part is only used for the final black box test. To better verify the efficacy of our SSEAT method, we also conduct more experiments over different datasets in the supplementary materials.

**Attack Algorithms.** Under the CDS task, we use various attack algorithms, *i.e.*, FGSM [19], BIM [30], PGD [40], RFGSM [53], MIM [13], NIM [36], SIM [36], DIM [60], VNIM [55], and VMIM [55], to generate adversarial samples under $l_\infty$ for training our SSEAT model and further evaluate the robustness against the attacks. The perturbation amplitude of all attacks is set to $\epsilon = 8/255$ and the attack step size to $\alpha = 2/255$. In our experiments, we conduct four different attack orders for the CDS task: (1) Order-I: FGSM, PGD, SIM, DIM, VNIM; (2) Order-II: BIM, RFGSM, MIM, NIM, VMIM; (3) Order-III: MIM, PGD, FGSM, SIM, BIM; (4) Order-IV: FGSM, BIM, PGD, RFGSM, NIM, SIM, DIM. We designed experiments involving

**Table 1: Comparing results of our SSEAT method with other adversarial training competitors under CDS task Order-I, including classification accuracy against attacks and standard accuracy on clean samples.**

| Method | FGSM | PGD | SIM | DIM | VNIM | Clean |
|---|---|---|---|---|---|---|
| PGD-AT[40] | 71.61 | 78.04 | 60.56 | 70.38 | 69.46 | 83.01 |
| TRADES[63] | 58.57 | 69.32 | 63.75 | 71.03 | 62.17 | 62.41 |
| MART [56] | 67.33 | 70.78 | 68.09 | 53.24 | 69.21 | 71.55 |
| AWP [59] | 49.99 | 68.75 | 59.93 | 70.74 | 44.88 | 79.67 |
| RNA [12] | 57.25 | 63.54 | 72.02 | 64.93 | 70.15 | 77.65 |
| LBGAT [10] | 65.35 | 73.19 | 75.54 | 68.38 | 72.37 | **83.22** |
| Baseline (CAD) | 68.21 | 72.49 | 73.12 | 71.18 | 73.35 | 60.52 |
| SSEAT (Ours) | **74.86** | **78.35** | **76.79** | **74.10** | **77.92** | 81.92 |

multiple attacks within a single stage. The sequence of attacks includes FGSM, BIM, PGD, RFGSM, NIM, SIM, partitioned based on encountering one, two, or three adversarial attacks per stage for training.

**Implementation Details.** We summarize the training procedure of our SSEAT framework in Alg. 1. The model uses resnet18 [20] as the classification network structure. We implement Torchattacks [25] to generate 100 adversarial images per category for each adversarial attack strategy, for a total of 1000 images. On the CIFAR10 dataset, before training, each image is resized to 32×32 pixels, and underwent data augmentation, which included horizontal flipping and random cropping. For the training hyperparameters of experiments, we use a batch size of 8, the epoch for training clean samples is set to 40, and the epoch for training adversarial samples is set to 20. We train the model using the SGD optimizer with momentum 0.9 and weight decay $5 \times 10^{-4}$. In addition, we set the memory buffer size to 1000.

**Competitors.** We compare our SSEAT method with several adversarial training works, such as the PGD-AT [40], TRADES[63], MART[56], AWP [59], RNA [12], and LBGAT [10].

**Evaluation Metrics.** To comprehensively assess the model's performance in both robustness and classification within the CDS task, we employ two indicators: (1) The model's classification accuracy across all kinds of adversarial examples after completing all adversarial training stages, which is more practical and differs from the 'classification accuracy against each attack after every adaptation step' [54]; (2) The model's classification accuracy on the original clean data after completing all adversarial training stages.

### 4.2 Experimental Results

We have conducted extensive experiments over the CIFAR-10 dataset with various attack orders in the CDS task, compared to several competitors and baseline. All attack results are reported under the black-box condition.

**For the CDS task, our SSEAT model can achieve the best robustness against ongoing generated new adversarial examples.** To better evaluate our SSEAT model under the CDS task, we generate 10 kinds of attacks and organize them as three different sets of adversarial sample sequences. We report model robustness over various attack orders compared to several competitors. As shown in Tab. 1, Tab. 2, and Tab. 3, our SSEAT method can beat

**Table 2: Comparing results of our SSEAT method with other adversarial training competitors under CDS task Order-II, including classification accuracy against attacks and standard accuracy on clean samples.**

| Method | BIM | RFGSM | MIM | NIM | VMIM | Clean |
|---|---|---|---|---|---|---|
| PGD-AT[40] | 71.63 | 69.04 | 72.87 | 63.90 | 67.55 | 83.01 |
| TRADES[63] | 70.26 | 68.84 | 62.49 | 59.37 | 67.92 | 62.41 |
| MART [56] | 68.78 | 53.68 | 70.84 | 69.47 | 63.25 | 71.55 |
| AWP [59] | 43.78 | 55.24 | 63.20 | 54.89 | 67.84 | 79.67 |
| RNA [12] | 67.51 | 70.55 | 68.52 | 69.56 | 58.26 | 77.65 |
| LBGAT [10] | 76.19 | 69.24 | 74.77 | 73.79 | 71.46 | **83.22** |
| Baseline (CAD) | 72.87 | 73.40 | 72.85 | 71.46 | 72.71 | 64.43 |
| SSEAT (Ours) | **76.77** | **77.01** | **77.48** | **76.62** | **75.51** | 82.79 |

**Table 3: Comparing results of our SSEAT method with other adversarial training competitors under CDS task Order-III, including classification accuracy against attacks and standard accuracy on clean samples.**

| Method | MIM | PGD | FGSM | SIM | BIM | Clean |
|---|---|---|---|---|---|---|
| PGD-AT[40] | 72.87 | 78.04 | 71.63 | 60.56 | 71.61 | 83.01 |
| TRADES[63] | 62.49 | 69.32 | 58.57 | 63.75 | 70.26 | 62.41 |
| MART [56] | 70.84 | 70.78 | 67.33 | 68.09 | 68.78 | 71.55 |
| AWP [59] | 63.20 | 68.75 | 67.33 | 59.93 | 43.78 | 79.67 |
| RNA [12] | 68.52 | 63.54 | 57.25 | 72.02 | 67.51 | 77.65 |
| LBGAT [10] | 74.77 | 73.19 | 65.35 | 75.54 | 76.19 | **83.22** |
| Baseline (CAD) | 71.73 | 73.67 | 70.38 | 69.52 | 72.67 | 63.39 |
| SSEAT (Ours) | **75.73** | **76.77** | **73.69** | **76.43** | **77.12** | 82.83 |

all adversarial training competitors for all adversarial examples, which demonstrate the efficacy of SSEAR framework to tackle the continuous new attacks under black-box condition.

**For the CDS task, our SSEAT model can maintain competitive classification accuracy over the clean data.** For real-life applications, in addition to continuously improving the defense performance against new attack methods, the model also needs to have good recognition effects on clean samples. Thus, we report the classification accuracy over original data in Tab. 1, Tab. 2, and Tab. 3. The results show our SSEAT method not only achieves strong robustness under complex changes, but also has good performance on the original task. Achieving a balanced trade-off between robustness and accuracy greatly enhances the practicality of the adversarial training strategy in real scenarios.

### 4.3 Ablation Study

To verify the role of each module in our SSEAT, we conducted extensive ablation studies on the following variants: (1) 'Baseline (CAD)': conduct the experiments over the continue learning pipeline; (2) '+KD': based on 'Baseline', add the knowledge distillation to shorten the distance between the defense current model and the clean model; (3) '+CRS': based on 'Baseline", add the CRS module for adversarial training under all stages; (4) '+Random DR': based on the "baseline", after each stage of training, a certain number of samples are

**Table 4: Results obtained from several variants of our SSEAT model.**

| Method | FGSM | BIM | PGD | RFGSM | NIM | Clean |
|---|---|---|---|---|---|---|
| Baseline (CAD) | 65.71 | 72.58 | 71.80 | 72.84 | 71.16 | 63.30 |
| +KD | 67.49 | 69.13 | 70.15 | 69.60 | 67.54 | 79.22 |
| +CRS | 70.78 | 73.97 | 73.89 | 73.20 | 71.96 | 80.22 |
| + Random DR | 72.02 | 75.16 | 75.36 | 75.14 | 73.73 | 81.35 |
| +ADR | 72.58 | 75.81 | 75.60 | 75.23 | 74.31 | 81.97 |
| +ADR+CRS (Simple) | 73.90 | 75.93 | 75.38 | 77.03 | 75.65 | 82.84 |
| +ADR+CRS (Ours) | **74.47** | **76.53** | **76.97** | **77.66** | **76.13** | **82.92** |

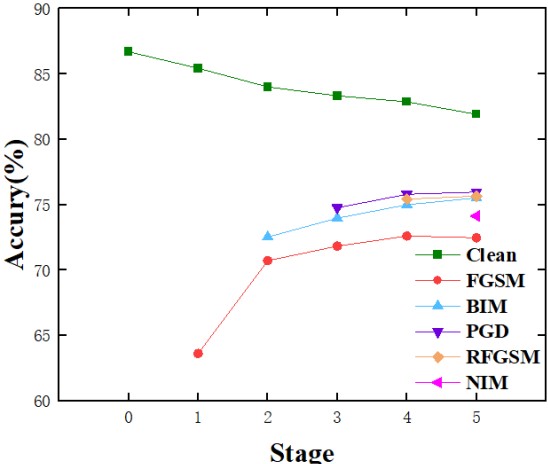

**Figure 3: Robustness accuracy of SSEAT model per training stage on CIFAR-10 for each attack method and clean samples.**

randomly selected and placed into the memory buffer for data reply; (5) '+ADR': based on the "baseline", add ADR for selecting key rehearsal data into the memory buffer; (6) '+ADR+CRS (Simple)': based on the 'baseline", ADR and CRS are added directly for model training, and CRS is used for all data during the training process; (7)'+ADR+CRS (Ours)': based on the 'baseline', add ADR and CRS for model learning with different training strategies (During the training process, the memory part is separated from the current data, and CRS is only used in the memory part). This is the overall framework of our SSEAT model.

**The efficacy of each component in the SSEAT method.** As shown in Tab. 4, by comparing the results of different variants, we notice the following observations, (1) 'Baseline (CAD)' based on the basic continue learning pipeline, can only realize learning from continuous attacks, however, the performance on clean samples is too poor to meet the actual application scenarios; (2) Comparing the results between '+Baseline (CAD)' and '+KD', we can notice the classification accuracy over the clean samples can be obviously improved; (3) Comparing the results between '+KD' and '+CRS', obviously, CRS is more suitable for the scenario of continuous adversarial samples. CRS can better handle the trade-off between model robustness and accuracy; (4) Comparing 'Baseline' and '+Random DR', the results prove the effectiveness of the knowledge replay

**Table 5: Comparing results of our SSEAT method with other adversarial training competitors under CDS task Order-IV, including classification accuracy against attacks and standard accuracy on clean samples.**

| Method | FGSM | BIM | PGD | RFGSM | NIM | SIM | DIM | Clean |
|---|---|---|---|---|---|---|---|---|
| PGD-AT[40] | 71.61 | 71.63 | 78.04 | 69.04 | 63.90 | 60.56 | 70.38 | 83.01 |
| TRADES[63] | 58.57 | 70.26 | 69.32 | 68.84 | 59.37 | 63.75 | 71.03 | 62.41 |
| MART [56] | 67.33 | 68.78 | 70.78 | 53.68 | 69.47 | 68.09 | 67.25 | 71.55 |
| AWP [59] | 49.99 | 43.78 | 68.75 | 55.24 | 54.89 | 59.93 | 53.24 | 79.67 |
| RNA [12] | 57.25 | 67.51 | 63.54 | 70.55 | 69.56 | 72.02 | 64.93 | 77.65 |
| LBGAT [10] | 65.35 | 76.19 | 73.19 | 69.24 | 73.79 | 75.54 | 68.38 | **83.22** |
| Baseline (CAD) | 63.84 | 72.28 | 73.15 | 72.39 | 70.81 | 72.29 | 71.65 | 60.78 |
| SSEAT (Ours) | **72.78** | **76.46** | **76.27** | **76.10** | **74.67** | **75.87** | **75.30** | 81.74 |

**Table 6: Comparing results of our SSEAT method with other adversarial training competitors under CDS task distributed Order-I, including classification accuracy against attacks and standard accuracy on clean samples.**

| Method | SIM | DIM | PGD | VNIM | FGSM | Clean |
|---|---|---|---|---|---|---|
| PGD-AT[40] | 60.56 | 70.38 | 78.04 | 69.46 | 71.61 | 83.01 |
| TRADES[63] | 63.75 | 71.03 | 69.32 | 62.17 | 58.57 | 62.41 |
| MART [56] | 68.09 | 53.24 | 70.78 | 69.21 | 67.33 | 71.55 |
| AWP [59] | 59.93 | 70.74 | 68.75 | 44.88 | 49.99 | 79.67 |
| RNA [12] | 72.02 | 64.93 | 63.54 | 70.15 | 57.25 | 77.65 |
| LBGAT [10] | 75.54 | 68.38 | 73.19 | 72.37 | 65.35 | **83.22** |
| Baseline (CAD) | 71.18 | 69.69 | 72.55 | 72.65 | 69.59 | 65.10 |
| SSEAT (Ours) | **75.72** | **72.72** | **76.74** | **74.04** | **73.98** | 82.46 |

**Table 7: The results of our SSEAT with multiple attacks in one training stage. 'Number', the first column in the table, represents the number of attack algorithms used in adversarial training in each stage.**

| attack | FGSM | BIM | PGD | RFGSM | NIM | SIM | clean |
|---|---|---|---|---|---|---|---|
| Number =1 | 72.78 | 76.06 | 76.27 | 76.10 | 74.67 | 74.37 | 81.74 |
| Number=2 | 73.62 | 76.11 | 76.76 | 76.42 | 74.54 | 74.74 | 82.75 |
| Number=3 | 73.72 | 76.66 | 76.88 | 76.48 | 74.65 | 74.44 | 83.47 |

strategy in the scenario of complex and diverse adversarial examples. Keeping some past data in memory can alleviate the model's loss of defensive performance against past adversarial examples; (5) Comparing the '+Random DR' and '+ADR', such results illustrate that our sampling strategy is effective, and can select representative and diverse samples at each training stage; (6) Comparing the '+CRS', '+ADR', and '+ADR+CRS (Simple)', the results demonstrate the ADR and CRS modules can complement each other on both classification accuracy and defense robustness; (7) Comparing '+ADR+CRS (Simple)' and '+ADR+CRS (Ours)', the results show our SSEAT model can achieve the best performance. The reason is that the model of the previous stage has knowledge of the data in memory, but does not have knowledge of the data in the current stage. If you move the model's output closer to it on the current data, it is likely to have a counterproductive effect and mislead the update of the current model; if you move the model's output closer to it on the memory data, it will guide the model to retain the knowledge of past samples. (8) As shown in Fig. 3, as training is completed, our SSEAT model can achieve a gradual improvement in defense performance and maintain classification accuracy.

**Our SSEAT model can be adapted to more challenging CDS tasks.** (1) Considering the complexity of real-world application scenarios, we extended the attack sequence to be longer. As shown in Tab. 5, thanks to the proposed data replay module and knowledge distillation strategy for alleviating the problem of catastrophic forgetting, our SSEAT model still shows the best defense and classification ability. (2) Additionally, we intentionally disrupted the original attack sequence to further validate the effectiveness of our method. As shown in Tab. 6, we show the results under re-ordering the attack sequence for Order-I in the CDS task, the defense performance still does not fluctuate excessively, which fully demonstrates that the SSEAT method can defend against a variety of adversarial samples. (3) Furthermore, to be more realistic, we designed a different setting in which the types of attack examples in each stage are not unique. As shown in Tab. 7, in the three stages of training with different numbers of types of attack samples, the model trained by our method maintains excellent accuracy on adversarial samples and clean samples. The results fully prove the versatility of our method and will not be affected by too many constraints. Overall, the biggest difference from other competitors is that our method model has the ability to evolve independently and can continue to learn and defend against more adversarial attacks. Our approach is able to adapt to complex and diverse attack samples and achieve broad and general model robustness.

## 5 CONCLUSION

To realize the adaptive defense ability of deep models with adversarial training when facing the continuously generated diverse new adversarial samples, we proposed a novel and task-driven Sustainable Self-evolution Adversarial Training (SSEAT) method. Inspired by the continue learning, the SSEAT framework can continuously learn new kinds of adversarial examples in each training stage, and realize the consolidation of old knowledge through the data rehearsal strategy of high-quality data selection. At the same time, a knowledge distillation strategy is used to further maintain model classification accuracy on clean samples. We have verified the SSEAT model efficacy over multiple continue defense setting orders, and the ablation experiments show the effectiveness of the components in the SSEAT method.

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
