# OpenReview forum: "Sustainable Self-evolution Adversarial Training"
_acmmm.org/ACMMM/2024/Conference — MM2024 Poster_

### Official Review · Reviewer_bGbz · 2024-05-24

**Rating:** 4
**Confidence:** 3

**Summary:**

This paper proposes a continual learning framework-based adversarial training method to guarantee the sustainable robustness of deep learning models toward novel adversarial attacks. The overall idea in this paper is simple and straightforwardly illustrated. However, the limitations of this paper include errors and ambiguities in notation, non-rigorous assertions, and the absence of somehow important experimental illustrations.

**Strengths:**

1. The problem related to the continuous robustness toward novel adversarial attacks is interesting and promising.

2. The core idea of this paper is straightforward and easy to understand.

3. The comparative experiment results in this paper are relatively sufficient.

**Limitations:**

1. There are some confused notations in section 3.2, especially with $P(y=c \mid x)$ from lines 462 - 485. In line 416, (x,y) is defined as an input-label pair with a certain mapping. I understand there is uncertainty from $x$ to $\tilde x$. Does it mean $P(f(g_k(x,\alpha _k))=y\mid x)$?.

2. The challenge mentioned in the introduction line 126 "How to balance the model’s robustness on adversarial examples and accuracy on clean data." is interesting. How to solve it by SSEAT? Could you give a brief explanation?

3.  A key problem SSEAT aims to solve is the robustness to tackle nove adversarial attacks. However, in the experimental part, from Tables 1 to 3, all attack algorithms are included in the training process of SSEAT. The performance of SSEAT under unknown adversarial attacks is important, yet this result is absent in the paper. Moreover, there is a small mistake in reporting the performance of PGD-AT under PGD attack. This method achieves SOTA performance but is not bold.

**Suitability:**

2

---

### Official Review · Reviewer_mjVG · 2024-05-24

**Rating:** 4
**Confidence:** 3

**Summary:**

The authors proposed an adversarial continual learning framework termed Sustainable Self-evolution Adversarial Training, which aims to improve defense performance for models in long-term applications.

**Strengths:**

This paper is well-written, and the training process is clearly explained.

This work is instructive as a robust model should have a long-term learning paradigm.

**Limitations:**

This work suggests that a robust model should also incorporate a continual learning framework to face growing malicious attacks. However, one difficulty in the traditional continual learning framework is handling new tasks or new categories, which may cause catastrophic forgetting.

In this work, using new attack methods as an analogy seems inappropriate. As far as current attack methods are concerned, the difference between the adversarial samples of new attacks and those of old attacks is not significant. For example, a model trained with samples generated by DI-FGSM can still defend against adversarial samples generated by PGD.

Additionally, setting the attack budget single to 8/255 makes it difficult to distinguish the disturbances caused by different attacks.

**Suitability:**

2

---

### Official Review · Reviewer_yuV7 · 2024-05-24

**Rating:** 4
**Confidence:** 3

**Summary:**

The paper introduces continual adversarial defense which combines continue learning and adversarial training to provide a novel Sustainable Self-evolution Adversarial Training framework. To incorporate continue learning into adversarial training, the authors introduce  consistency regularization strategy and adversarial data replay to retain past knowledge and tackle model catastrophic forgetting. The experiments demonstrate the effectiveness of the proposed algorithm.

**Strengths:**

1. The proposed consistency regularization strategy and adversarial data replay seem natural for the combination of continue learning and adversarial training.
2. The experiment results demonstrate the superiority of the proposed algorithm over other baselines.
3. The ablation studies demonstrate the effectiveness of each component.

**Limitations:**

1. There is no evaluation of different attack strengths, such as PGD attacks with more attack iterations and larger perturbation sizes.
2. There is no evaluation of black-box attacks.
3. More evaluation on multiple datasets is recommended.

**Suitability:**

2

---

### Official Review · Reviewer_4Ry7 · 2024-05-27

**Rating:** 2
**Confidence:** 3

**Summary:**

This paper extends the adversarial training in the continuous learning setting. The proposed method, called SSEAT, involves an adversarial data reply module to select relearning data and a consistency regularization to retain past knowledge and maintain accuracy on clean data. Experiments on CIFAR-10 dataset demonstrate superior defense performance over the baselines.

**Strengths:**

-Exploring model robustness in the long-term learning process is encouraged.

-The paper is easy to read.

**Limitations:**

This paper explores continuous adversarial defense in multiple-stage learning, which is interesting and encouraged. However, the effect of the proposed method is limited and the experiments are insufficient.

-The adversarial examples in the paper are all gradient-based. Since there exist many adversarial attack methods, i.e., learning-based and optimization-based, they should be also explored in the continuous defense.

-The experiments are only conducted on CIFAR-10 dataset. More complex datasets should be tested.

-Most of the compared methods are one-time learning, applying different training methods in the CAD setting is more appropriate.

-I'm curious about the distribution of the adversarial examples selected in the replay memory module. Since the adversarial examples are all gradient-based, would the strongest adversarial examples (PGD) always be selected?

**Suitability:**

3

---

### Meta-Review · Area_Chair_TzwX · 2024-07-02

**Recommendation:** Accept (Poster)
**Confidence:** 3

**Metareview:**

The review comments are only lukewarm. Some reviewers complained about inadequacy of the rebuttal in addressing the original issues.

The proposed approach has some merits, but the issues probably outweigh the strengths. So I would vote for Reject, but will be open to a different outcome.